# Effect of Serious Gaming on Speech-in-Noise Intelligibility in Adult Cochlear Implantees: A Randomized Controlled Study

**DOI:** 10.3390/jcm11102880

**Published:** 2022-05-19

**Authors:** Pierre Reynard, Virginie Attina, Samar Idriss, Ruben Hermann, Claire Barilly, Evelyne Veuillet, Charles-Alexandre Joly, Hung Thai-Van

**Affiliations:** 1Institut de l’Audition, Institut Pasteur, Université de Paris, INSERM, 75012 Paris, France; pierre.reynard@hotmail.fr (P.R.); evelyne.veuillet@gmail.com (E.V.); charles-alexandre.joly01@chu-lyon.fr (C.-A.J.); 2Faculty of Medicine, University Claude Bernard Lyon 1, 69100 Villeurbanne, France; ruben.hermann@chu-lyon.fr; 3Hospices Civils de Lyon, Hôpital Edouard Herriot, Service d’Audiologie et Explorations Otoneurologiques, 69003 Lyon, France; virginie.attina@gmail.com (V.A.); samar.a.idriss@hotmail.com (S.I.); 4Lyon Neuroscience Research Center, INSERM U1028, CNRS UMR5292, Integrative, Multisensory, Perception, Action and Cognition Team (IMPACT), 69675 Bron, France; 5Hospices Civils de Lyon, Hôpital Edouard Herriot, Service d’ORL, Chirurgie Cervico-Faciale et d’Audiophonologie, 69003 Lyon, France; claire.barilly@gmail.com

**Keywords:** serious game, auditory rehabilitation, cochlear implant, listening-in-noise, speech reception threshold, re-test

## Abstract

Listening in noise remains challenging for adults with cochlear implants (CI) even after prolonged experience. Personalized auditory training (AT) programs can be proposed to improve specific auditory skills in adults with CI. The objective of this study was to assess serious gaming as a rehabilitation tool to improve speech-in-noise intelligibility in adult CI users. Thirty subjects with bilateral profound hearing loss and at least 9 months of CI experience were randomized to participate in a 5-week serious game-based AT program (*n* = 15) or a control group (*n* = 15). All participants were tested at enrolment and at 5 weeks using the sentence recognition-in-noise matrix test to measure the signal-to-noise ratio (SNR) allowing 70% of speech-in-noise understanding (70% speech reception threshold, SRT70). Thirteen subjects completed the AT program and nine of them were re-tested 5 weeks later. The mean SRT70 improved from 15.5 dB to 11.5 dB SNR after 5 weeks of AT (*p* < 0.001). No significant change in SRT70 was observed in the control group. In the study group, the magnitude of SRT70 improvement was not correlated to the total number of AT hours. A large inter-patient variability was observed for speech-in-noise intelligibility measured once the AT program was completed and at re-test. The results suggest that serious game-based AT may improve speech-in-noise intelligibility in adult CI users. Potential sources of inter-patient variability are discussed. Serious gaming may be considered as a complementary training approach for improving CI outcomes in adults.

## 1. Introduction

Since the early 1990s, cochlear implants (CI) have undoubtedly provided improvements in terms of the quality of life and auditory skills of both adults and children. However, some limitations remain [1]. Immediately after CI surgery, patients must adapt to perceiving new sounds, which they learn to recognize with the assistance of speech therapy. CI recipients need to learn how to treat sound flow and to mentally represent the relationships between the perceived sounds (signifier) and their meaning (signified) to improve their auditory skills.

Auditory training (AT) has been used since the early 1970s to teach a wide range of auditory skills, including detection (i.e., to be aware of the absence or presence of a target sound-alert function), discrimination (i.e., to distinguish between sounds), identification (i.e., to identify words, pseudo-words, syllables, phonemes), and comprehension (i.e., to make sense of the sounds heard, whether they are environmental (noise) or linguistic). In CI recipients, there is sparse evidence on the efficacy of AT, possibly due to the heterogeneity of training protocols, outcome measures, and demographic data [2].

Understanding in noise and suprasegmental speech parameter perception and interpretation (i.e., recognizing prosodic variations, rhythms, intonations) remain crucial in AT. The latter must focus on both verbal working memory abilities, and executive functions, such as attention (alertness, sustained attention, selective attention) and inhibition. Studies have found a correlation between verbal working memory abilities and speech comprehension in noise, meaning that knowledge and neurocognitive functions may influence the results of speech-in-noise intelligibility [2,3,4]. 

Speech recognition in a noisy environment is challenging for CI recipients, even for those with prolonged experience: speech recognition in CI listeners is more impaired by background noise than that of normal-hearing (NH) listeners [5]. Compared to NH listeners, CI recipients need a signal-to-noise ratio (SNR) at least 25 dB higher than NH listeners to reach the 50% speech reception threshold (SRT50), i.e., to be able to repeat 50% of the linguistic material delivered in the presence of noise [6]. As expected, speech recognition and sound localization in noisy environments is better in bilateral CI users compared to unilateral users [7,8]. Although AT has previously been reported to improve speech-in-noise intelligibility in subjects with hearing aids [9,10], this result is still debated. For instance, when Abrams et al. investigated the effect of computer-assisted AT (CAAT) on the listening skills in noise of a sample of subjects with newly fitted hearing aids, the authors found no significant improvement, which they believed was due to difficulties related to program compliance [11].

Despite technological advances, CI alone do not enable the satisfactory restoration of auditory skills and there is a consensus that speech re-education or AT is essential [12,13,14,15]. Traditionally, AT is provided in a face-to-face setting; however, there are some reports of computerized AT (CAT) programs for adult CI recipients, but not all are based on serious gaming [1,16,17,18]. AT programs can now be followed remotely, via computer or mobile applications [19,20]. The objective of AT is to stimulate the plasticity of rehabilitation, and research has shown that neurophysiological changes can occur after the placement of CI [21]. After activation of the implant, active rehabilitation strategies, based on explicit AT, show better results than passive strategies [22]. The period of auditory adaptation to ensure good post-implantation results varies for adult CI recipients. However, not all implanted subjects are offered active AT, not only because of its cost and the lack of speech therapists, but also due to the lack of consensus concerning therapeutic strategies [22]. It is, however, increasingly recognized that subjects need to be more involved in their aural rehabilitative process and that more options to personalize their rehabilitative program should be offered [23].

Serious gaming is an emerging applied field of research that focuses on the use of digital gaming platforms and technologies for more than just entertainment [24,25]. One suggested definition is “a mental contest, played with a computer in accordance with specific rules that uses entertainment to further government or corporate training, education, health, public policy, and strategic communication objectives” [26]. Serious games have been used in a variety of fields such as education, asthma education, psychotherapy, and even surgical training [27,28,29,30,31]. By offering a pleasant game experience, the use of serious game-based training is thought to significantly boost interest and motivation and thus reinforce the players’ acquisitions in the trained domain [32]. Serious game-based programs may be adapted to the training needs specifically met by CI users.

To date, no study has evaluated the value of serious game-based AT in CI subjects. As speech comprehension in competitive listening situations remains a challenging improvement goal in CI adults, evaluating the effect of serious game-based AT on speech in noise intelligibility in this population is of great interest. The primary objective of the present study was to evaluate the efficacy of a 5-week digital gaming program in this regard. The secondary objective was to evaluate the maintenance of possible benefits over time.

## 2. Materials and Methods

### 2.1. Participants

A total of 30 adults with at least 9 months of CI experience were recruited at the department of audiology and otoneurology of the Edouard Herriot University Hospital, Lyon, France (Figure 1).

Eligible subjects were over 18 years old, suffered from bilateral profound hearing loss, and had had unilateral or bilateral CI for at least 9 months (range 1 to 26 years). All participants reported auditory difficulties in a noisy environment. The study protocol was approved by the local ethics committee (CPP Sud-Est IV 14/034 ID RCB 2014-A00345-42). Written informed consent was obtained from all patients. 

The CI subjects were randomized into two groups using a computer-generated randomization list: an intervention group, which was called the study group (*n* = 15, 7 males, 8 females; mean age, 48 years, range 24 to 76 years) and an untrained group (control group) (*n* = 15, 8 males, 7 females; mean age 60 years, range 45 to 75 years). None of the subjects followed any other AT program during the study.

### 2.2. Intervention

With the aim of providing innovative and translational therapeutic methods in CI adults, a dedicated serious game was developed with the support of the French government (“Neurosyllabic R&D project”). The design and development of the serious game were based on previously published criteria for an effective AT protocol [10]. These criteria included ease of access (achievable at home and suitable for the elderly), interactivity, tasks of increasing complexity (to maintain the interest and attention of the subject), feedback, and the ability to record performances at any time.

A simple serious game scenario was developed in order to enable most subjects to easily identify with an avatar (Figure 2). Participants underwent a 5-week training program including 6 activities. The first 2 consisted in detecting and discriminating target sounds (animal calls, instruments, everyday noises, and words) in noise. These 2 activities were the only ones available during the 1st week. Then, 4 other games were introduced in the 2nd week: 1 consisted in target sound identification, and the last 3 were word-based games during which the subject had to either discriminate words, identify their syllables, or categorize them according to their semantic. 

The auditory material included 240 noises, 22 instrument sounds, 100 animal calls, 3135 words, 665 logatomes, and 600 syllables, while the video material contained 1400 illustrative images. Among all the sounds, syllables, and words used, 30% were selected from a dedicated database created for the study, 40% were recorded by professional actors, and 30% (especially ambient background sounds) were purchased from a database on the Internet. In order for training to remain close to real-life conditions, while allowing a progressive increase in difficulty, for the first 2 games, subjects could choose from 4 types of ambient sounds each of which had a variable signal-to-noise ratio (SNR): white noise, continuous noises (sound of rain, wind, etc.), discontinuous noises (such as the auditory environments of everyday life), or babbling noises.

The game was automatically adapted in terms of difficulty. The volumes of the target sounds and the ambient sound (SNR) were adjusted according to 20 levels of difficulty. For levels 1 to 10, the target sounds were set at 100%, while the volume of the ambient sound increased from 0 to 90%. For levels 11 to 20, the ambient sound was set at 100%, while the volume of the target sounds decreased from 100% to 10%. The level of difficulty could either be set manually (in which case, each game had a fixed duration of 2 min) or adapted automatically by an algorithm (the game then stopped after 4 errors). In case of automatic management, the level of difficulty was set according to the previous games: it increased after each correct answer and decreased after each error. Adaptive changes in the difficulty level depended on 3 factors:

The probability of reaching a correct answer by chance (for instance, the increment in difficulty was lower if there was 1 correct answer among 2 than if there was 1 among 5).

Elapsed time: the more time passed, the greater the increment in difficulty and the smaller the decrement. This ensured that each game did not last too long.

The number of errors and correct answers that already occurred. A sequence of several mistakes without any correct answer since the beginning of the game meant that the initial level of difficulty was too high and therefore needed to be adjusted more quickly. Conversely, a faultless course led to a faster increase in difficulty.

### 2.3. Experimental Protocol

The study group was instructed to undergo a minimum of 20 training sessions over a period of 5 weeks. One of the weekly sessions was performed at the hospital under the supervision of a board-certified audiologist. During the hospital session, the serious game parameters were constant, except for difficulty, which was increased as the patient progressed. The parameters of the 6 activities were unchanged. At home, the subjects carried out the other sessions by logging onto an online platform using their personal identifier. To ensure the regularity of the training, the home sessions were remotely controlled. Subjects were advised to sit comfortably in a quiet room; the noise level at the beginning of the game session was adjustable. As the speakers were often integrated into their computers, no further instructions regarding speaker placement were given. During the hospital sessions, the duration of each game was set at 2 min and the experimenter set the initial difficulty level (SNR) of tasks 1 and 2. In order to ensure that the level of difficulty was appropriate, the difficulty was determined automatically via an adaptive algorithm. To maintain a high level of motivation during the training sessions at home, the duration of the games could vary according to the performance of the participants. For each activity, gaming stopped as soon as the subject made 4 mistakes. 

### 2.4. Data Logging

For each exercise carried out, the date, the total duration, and the actual playing duration were gathered on the online platform. This enabled the total number of exercises and the total playing time of all participants to be recorded.

### 2.5. Pre- and Post-Auditory Training Assessment of Speech-in-Noise

A pre- and post-AT assessment was conducted at enrolment (T1) and 5 weeks later (T2) using speech-in-noise audiometry for all participants. Additionally, 9 subjects from the study group agreed to be re-tested 5 weeks after the training period (T3) to evaluate if the benefit was maintained over time. 

To assess speech-in-noise before and after training, the French version of the matrix test (Fr-matrix; adaptive procedure; system Ear 3.0, Auritec, Hamburg, Germany) was used since it exhibits high discriminative power, both in stationary and in fluctuating noise settings [33]. In this test, the speech reception threshold (SRT), which is the stimulus presentation level (relative to the noise level), is usually set to a recognition score of 50% (normative value: SRT 50 = −6.0 ± 0.6 dB SNR). The stimuli library contained 50 French words (10 names, 10 verbs, 10 numerals, 10 objects, and 10 colors) that were selected based on their phonetic content to represent the mean phonetic distribution in French spoken language. An advantage of this tool is the absence of any learning effect, which is particularly useful for repeated assessments [34].

Herein, following national guidelines for speech-in-noise testing in adults [35], the target threshold was fixed at 70% (SRT70) on purpose to avoid subjects experiencing a feeling of early failure, and was measured at T1, T2, and T3. To do so, 2 lists of words in a silent condition (20 randomly generated sentences) and 3 other lists with background noise (steady intensity of 60 dB) were played via 2 loudspeakers positioned 1 m in front of the patient in a soundproof booth. The examiner, a board-certified audiologist, was seated next to the patient in the booth. 

The subjects in the study group underwent a semi-structured interview after the end of the training. They were asked: “Did you enjoy the training program?” and “Did the training improve your listening-in-noise skills?”.

### 2.6. Statistical Analysis

Statistical analyses were performed using the SigmaStat^®^ software (Systat Software, San Jose, CA, USA) and R version 4.1.2 (R Core Team 2021, R Foundation for Statistical Computing, Vienna, Austria). As they followed a normal distribution (confirmed by a Kolmogorov–Smirnov test), SRT70 values measured at T1 were compared between groups using a *t*-test. In each group, SRT70 values measured at T2 were compared to T1 values using paired *t*-tests. 

To control for potential differences in demographics (age at testing, deafness duration prior to implant, years of implant experience) between groups, the *t*-test and Wilcoxon test were used. A possible correlation between demographics and SRT70 improvement between T1 and T2 was also tested.

In contrast, the total number of games played and the total duration of play were not normally distributed. The correlations of these 2 variables with each other and with SRT70 changes as a result of training were assessed using Spearman’s correlation tests.

## 3. Results

Patient characteristics are summarized in Table 1.

Among all participants, two from the study group did not complete the training and were excluded (one moved, the other gave up), leaving 13/15 subjects (87%) who completed training and post-training Fr-matrix assessments. The time spent playing varied between 4 h 24 min and 39 h (mean 13 h) for a total of 141 to 973 exercises performed (mean 368); the number of games played was significantly correlated with the duration of play (Spearman rho = 0.951; *p* < 0.001).

Before the intervention, the initial results from the Fr-matrix assessments were not significantly different between the study and control groups (t = 0.688 with 26 degrees of freedom; *p* = 0.49). Mean age differed between the study and control groups (*t*-test, *p* = 0.039). Age at testing, however, was not correlated with SRT70 improvement between T1 and T2 (Pearson test, *p* = 0.525). Moreover, neither deafness duration prior to implant (*t*-test, *p* = 0.449) nor the number of years of implant experience (Wilcoxon test, *p* = 0.487) differed between groups. Further, SRT70 improvement between T1 and T2 did not correlate with deafness duration (Pearson test, *p* = 0.071) nor with CI experience (Spearman test, *p* = 0.360).

In the control group, the mean difference in SRT70 between T1 (12.66 dB) and T2 (11.60 dB) was not significant (t14df-test = 0.655; *p* = 0.523, Table 2).

In the study group, a significant difference in speech-in-noise intelligibility was found between pre- and post-test assessments. The mean SRT70 in the study group was 15.5 dB at T1, and 11.5 dB at T2 (t12df-test = 4.521; *p* < 0.001; Figure 3). The mean SNR gain at SRT70 was −3.98 dB, with 6 of the 13 subjects evaluated having gained at least −4 dB SNR (Median = −2.8 dB SNR). All trained subjects improved their hearing abilities in noise, with decreased SRT70 after training, except Patient 5 (a 70-year-old male with 1 year of CI experience) whose SRT70 remained stable post-training (Table 2; Figure 3). The largest reduction in SRT70 was −10.2 dB SNR (Patient 12). Changes in SRT70 between T1 and T2 were not correlated with the number of games played (Spearman rho = −0.130; *p* = 0.693) nor with the total duration of play (Spearman rho = 0.033; *p* = 0.915).

All 13 participants in the study group responded ‘Yes’ to the two questions in the exit interview, i.e., “Did you enjoy the training program?” and “Did the training improve your listening-in-noise skills?”.

At T3, eight out of the nine re-tested subjects still presented a decrease in SRT70 compared to T1, and the mean difference between T1 and T3 was of −2.28 dB. The mean SRT70 difference between T2 and T3 was +1.13, ranging from −4.0 in Patient 9 to +10.2 in Patient 8. Only one patient (Patient 5) did not show an overall improvement between T1 and T3 (Figure 4).

## 4. Discussion

This study provides evidence of the impact of serious gaming on speech-in-noise intelligibility in adult CI users.

The Fr-matrix SRT70 was used as a measure of speech-in-noise intelligibility for assessing the effectiveness of a 5-week AT and its persistence. To remain as close as possible to real-life listening situations, the training assessment was performed using sentences and informational masking noise. Our group previously reported that, among speech-in-noise tests suitable for French-speaking populations, the Fr-matrix provides the lowest intra-subject variability (±0.6 dB for SRT50) [34,35].

Herein, the post-training improvement in SRT70 was measured at a mean of −3.98 dB, a result that cannot be attributed to either intra-individual variation or to procedural learning alone. The latter is, in fact, evaluated at 1.8 dB for the Fr-matrix test [33]. Moreover, the improvement in SRT70 was observed in 12 of the 13 trained subjects. In the patient who did not improve, the SRT70 degradation was minimal (+0.1 dB SNR). Conversely, the control group did not show an overall improvement. More precisely, eight subjects from the control group showed an improvement in SRT70 ranging from −0.9 to −13.3 dB SNR (mean −5.2 dB SNR), while seven showed a degradation ranging from +0.5 to +14.6 dB SNR (mean +3.6 dB SNR). Even when excluding the control patient with the highest SRT70 degradation after 5 weeks (+14.6 dB SNR), the mean SRT70 values after 5 weeks were still not significantly different from those measured initially (t13df = 1.733; *p* = 0.107). Among the participants’ demographic characteristics, only mean age differed between the study and control groups. None of the demographic characteristics, including deafness duration and experience with the implant, were found to correlate with improvement in SRT70.

In the nine subjects of the study group re-tested 5 weeks after the end of the intervention, only one had a worse SRT70 than before training (difference T3-T1 = +6.2 dB SNR). For the other eight patients, the SRT70 remained better than before training: three subjects had a gain of between −0.5 and −2 dB SNR and five maintained a gain of greater than −3 dB. However, the mean difference in SRT70 (−2.28 dB SNR) measured between inclusion and re-test at 10 weeks was not significant. To date, only one study has measured the persistence of the efficacy of computer-assisted AT on speech-in-noise intelligibility in CI users [16]. These authors showed that, in 10 adult CI subjects, the benefit of AT on SRT50 could be observed up to 4 weeks after the end of the training with a gain of 2dB SNR. Future studies should more systematically integrate follow-up evaluation sessions to assess the long-term benefits of AT [36].

The serious game we used was developed specifically for this study. The software and its content had not been subject to a previous validation study. During the procedure, participants performed one training session face-to-face in the laboratory each week to ensure that the game’s instructions were understood and well-followed during training, and to collect the user’s experience over the previous week. The rest of the training was carried out remotely via the online gaming platform. In order to preserve the playful nature of AT, the duration of the training, the choice of activities among the six available options, and the initial difficulty level were left to the participant’s will. However, an adaptive training procedure was used, in order to minimize the potential effect of inter-individual differences in initial SNR values.

Each participant was instructed to do a minimum of four training sessions per week, which was the case for each of them. The number of games played per session, however, was left up to the players in order to encourage their adherence. The relationship between the magnitude of improvement and the cumulative duration, in hours, over the 5 weeks of training could be assessed, since training logs were collected. Although the duration of training was highly variable between subjects, it was not associated with SRT70 improvement. The patient who participated the most showed an improvement at T2 (−6.9 dB SNR) compared to T1, which was higher than the mean SRT70 improvement. However, other subjects with less total training time (Patients 1, 10, and 12) showed a higher improvement (−7.9, −7.4, and −10.2, respectively) even though they had completed fewer games than the mean number of games played (336, 162, and 141, respectively). Furthermore, the patient with the highest improvement was the one who played the least. This result indicates that, while training had an overall beneficial effect and was measurable in almost all participants, there were large inter-individual disparities in the magnitude of SRT70 improvement, which prevailed over the total training time. While a weekly training schedule was set in the present study, only one study, to our knowledge, has evaluated the impact of AT schedule on speech recognition performance in degraded listening situations [37]. By training NH adults to recognize modulated vowels via a CI simulator, the authors did not find any influence of the pace of the training sessions on recognition improvement.

All or part of the inter-individual variability observed in speech-in-noise intelligibility improvement could be due to differences in the supraliminal abilities of the participants. Meta-analyses conducted in adult CI users provided evidence that demographic factors such as deafness duration or age at onset were predictive of CI outcomes, although they only explained 20% of the variance [38,39,40]. Furthermore, the sole SRT70 as a supraliminal measure does not account entirely for the patient’s ability to recognize speech in noise. A recent meta-analysis identified the involvement of particular cognitive domains associated with speech-in-noise intelligibility, namely, processing speed, inhibitory control, working and episodic memory, and crystallized intelligence [41]. However, taken together, these cognitive abilities explain less than 10% of the inter-individual variability. A more recent review underlined the relationship between profound deafness of genetic origin and the occurrence of central auditory processing disorders in mice [42]. This is in full agreement with the fact that for a given degree of hearing loss, supraliminal auditory performance may considerably vary from one subject to another. Further studies on serious game-based AT are needed in order to better control cognitive biases potentially affecting speech comprehension in noise.

## Figures and Tables

**Figure 1 jcm-11-02880-f001:**
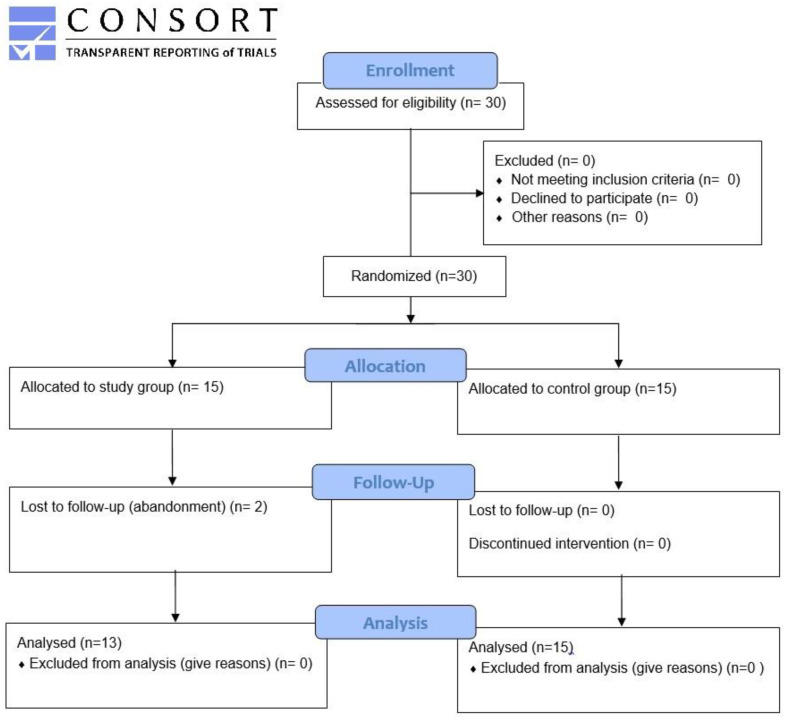
Flow chart.

**Figure 2 jcm-11-02880-f002:**
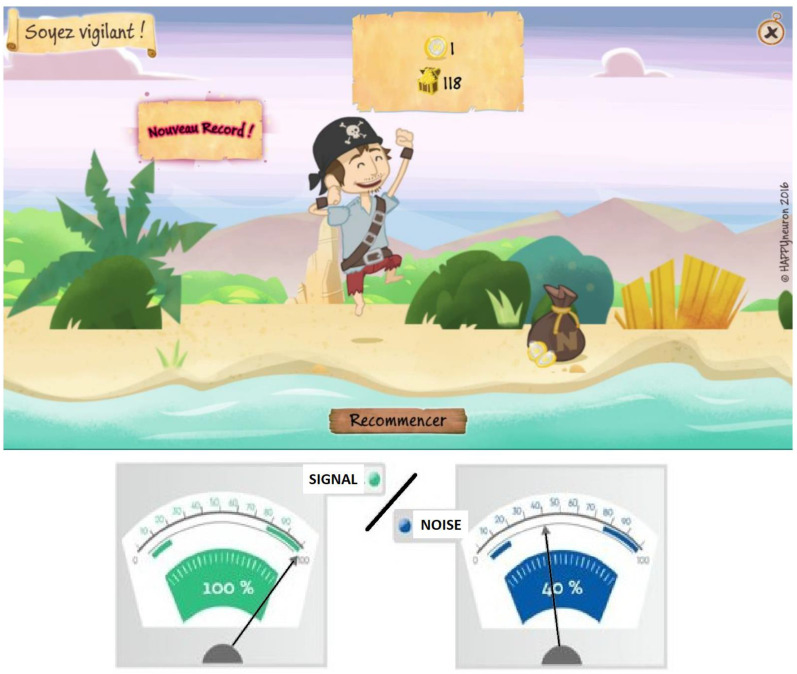
Serious game visuals with signal-to-noise ratio representation. As the player is detecting or identifying target sounds in the presence of background noise, the avatar is walking along a beach to collect coins. For each incorrect answer, the avatar falls and slightly regresses. After 4 incorrect answers or a pre-set time has elapsed, the game stops. The player is expected to collect as many coins as possible in 1 game with an updated score available on the screen at the end of each game. This playful mechanism encourages the player to immediately play again in an attempt to beat his/her personal record.

**Figure 3 jcm-11-02880-f003:**
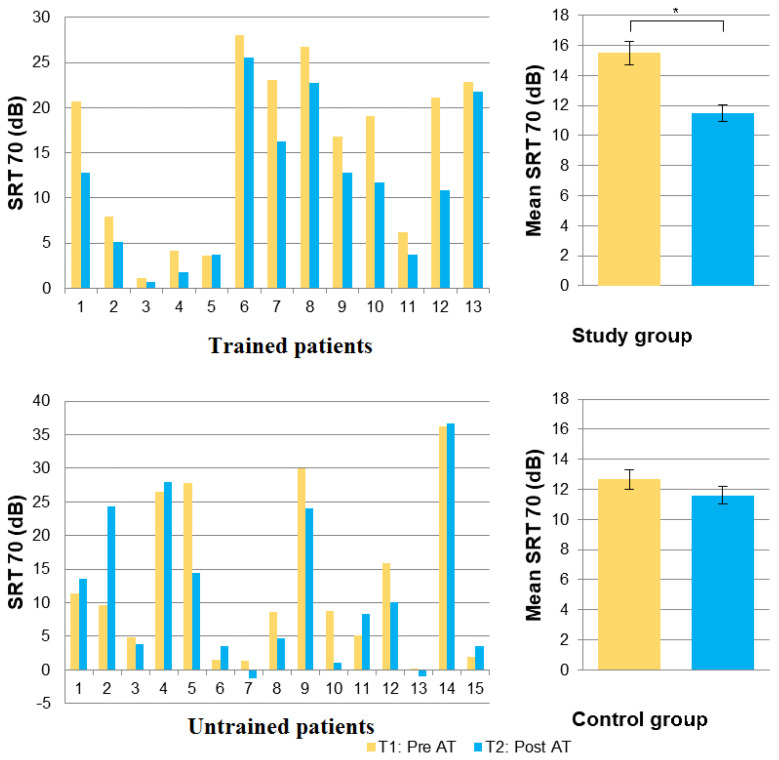
Changes over time in signal-to-noise ratio (dB) at a 70% speech reception threshold. Individual results are shown on the left and mean group results on the right in the study group (**top panel**) and control group (**bottom panel**); testing at enrollment (yellow) and at 5 weeks (blue). The difference is significant only in the study group (noted *).

**Figure 4 jcm-11-02880-f004:**
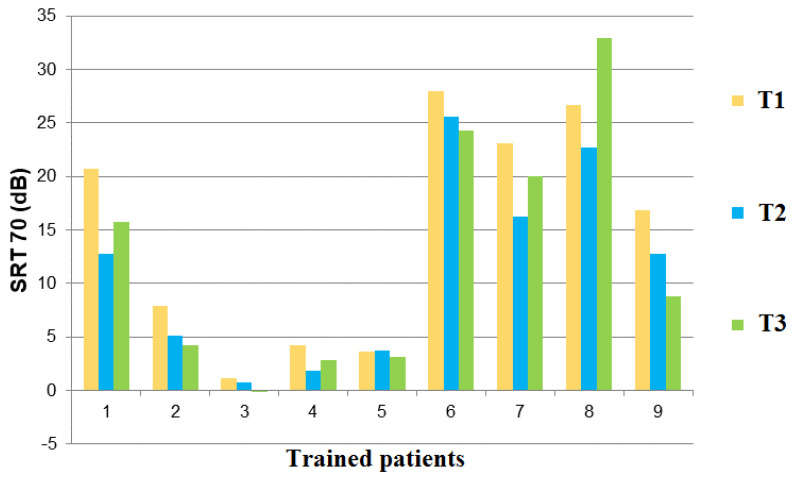
Changes over time in signal-to-noise ratio (dB) at a 70% speech reception threshold for nine subjects of the study group, 5 weeks after serious game-based AT (green).

**Table 1 jcm-11-02880-t001:** Demographic data for trained and untrained participants (CI = cochlear implant; HA = hearing aid; RE = right ear; LE = left ear; SNHL = sensori-neural hearing loss).

Patient	Age (Years)	Sex	Deafness Duration (Years)	Deafness Etiology	CI Experience (Years)	Side of CI and HA	CI Manufacturer
** *Study group* **							
1	38	M	35	Progressive SNHL	3	CI: RE/CI: LE	Oticon Medical/Neurelec Digisonic SP
2	76	M	26	Presbycusis	5	CI: RE	Oticon Medical/Neurelec Digisonic SP
3	31	F	28	Meningitis	6	CI: LE/HA: RE	Oticon Medical/Neurelec Digisonic SP
4	56	F	26	Otosclerosis	3	CI: RE/CI: LE	Cochlear
5	70	M	15	Otosclerosis	1	CI: LE/HA: RE	AB Naida CI Q70
6	29	M	28	Meningitis	26	CI: RE/CI: LE	Cochlear
7	35	M	35	Progressive SNHL	1	CI: LE/HA: RE	AB Naida CI Q70
8	46	F	20	Progressive SNHL	7	CI: RE/CI: LE	Medel Concerto
9	76	F	16	Presbycusis	2	CI: RE/HA: LE	Medel Concerto
10	69	M	19	Otosclerosis	14	CI: RE	Neurelec
11	71	F	21	Presbycusis	1	CI: RE/HA: LE	Oticon Medical/Neurelec Digisonic SP
12	44	F	5	Meningitis	5	CI: RE	Medel Concerto
13	25	F	25	Genetic	19	CI: RE	Cochlear
14	37	F	36	Genetic	25	CI: RE/CI: LE	AB Naida CI Q70
15	24	M	24	Genetic	13	CI: RE/CI: LE	Cochlear
** *Control group* **							
1	75	F	25	Progressive SNHL	7	CI: LE/HA: RE	Oticon Medical/Neurelec Digisonic SP
2	67	F	17	Progressive SNHL	3	CI: RE/HA: LE	Oticon Medical/Neurelec Digisonic SP
3	63	M	20	Otosclerosis	4	CI: RE/CI: LE	Oticon Medical/Neurelec Digisonic SP
4	45	M	39	Progressive SNHL	4	CI: RE	Oticon Medical/Neurelec Digisonic SP
5	68	M	5	Traumatic	4	CI: RE/CI: LE	Oticon Medical/Neurelec Digisonic SP
6	49	F	25	Genetic	6	CI: LE/HA: RE	Medel Concerto
7	55	F	30	Meningitis	5	CI: RE	Oticon Medical/Neurelec Digisonic SP
8	67	F	16	Progressive SNHL	8	CI: RE	Medel Concerto
9	67	M	15	Otosclerosis	8	CI: RE/CI: LE	Oticon Medical/Neurelec Digisonic SP
10	46	M	16	Iatrogenic	3	CI: LE/HA: RE	Cochlear
11	53	M	40	Genetic	9	CI: RE/CI: LE	Oticon Medical/Neurelec Digisonic SP
12	59	M	20	Menière	2	CI: LE/HA: RE	Medel Concerto
13	58	F	50	Genetic	19	CI: RE/CI: LE	Cochlear
14	73	M	23	Presbycusis	3	CI: RE/CI: LE	Oticon Medical/Neurelec Digisonic SP
15	63	F	55	Genetic	9	CI: RE/CI: LE	Oticon Medical

**Table 2 jcm-11-02880-t002:** Individual and mean signal-to-noise ratio (SNR) results from Fr-matrix for the study and control groups at enrollment (T1), at 5 weeks (T2), and, for the study group, 5 weeks post-intervention (T3).

Signal-to-Noise Ratio (dB) (Fr-Matrix Results)						
** *Study group* **						
**Patient**	**T1**	**T2**	**Δ T2−T1**	**T3**	**Δ T3−T1**	**Δ T3−T2**
1	20.7	12.8	−7.9	15.7	−5.0	+2.9
2	7.9	5.1	−2.8	4.2	−3.7	−0.9
3	1.1	0.7	−0.4	−0.2	−1.3	−0.9
4	4.2	1.8	−2.4	2.8	−1.4	+1
5	3.6	3.7	+0.1	3.1	−0.5	−0.6
6	28.0	25.6	−2.4	24.3	−3.7	−1.3
7	23.1	16.2	−6.9	20.0	−3.1	+3.8
8	26.7	22.7	−4.0	32.9	+6.2	+10.2
9	16.8	12.8	−4.0	8.8	−8.0	−4.0
10	19.1	11.7	−7.4	NA		
11	6.2	3.7	−2.5	NA		
12	21.1	10.9	−10.2	NA		
13	22.8	21.8	−1.0	NA		
**Mean**	**15.48**	**11.50**	**−3.98**	**12.40**	**−2.28**	**+1.13**
**SD**	**9.52**	**8.31**		**11.45**		
**SEM**	**2.64**	**2.71**				
** *Control group* **						
**Signal-to-noise ratio (dB) (Fr-matrix results)**						
**Patient**	**T1**	**T2**	**Δ T2−T1**			
1	11.4	13.5	+2.1			
2	9.7	24.3	+14.6			
3	4.8	3.9	−0.9			
4	26.5	28	+1.5			
5	27.8	14.5	−13.3			
6	1.5	3.5	+2			
7	1.4	−1.2	−2.6			
8	8.6	4.7	−3.9			
9	30	24	−6			
10	8.8	1	−7.8			
11	5.2	8.3	+3.1			
12	15.9	10.1	−5.8			
13	0.1	−0.9	−1			
14	36.2	36.7	+0.5			
15	2	3.6	+1.6			
**Mean**	**12.66**	**11.6**	**−1.06**			
**SD**	**11.86**	**11.67**				
**SEM**	**3.04**	**2.99**				

## Data Availability

The data presented in this study are available on request from the corresponding author. The data are not publicly available due to ethical, legal and privacy issues.

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
