# Peer review of "Effect of Serious Gaming on Speech-in-Noise Intelligibility in Adult Cochlear Implantees: A Randomized Controlled Study"

_jcm, 2022, doi:10.3390/jcm11102880_

Round 1

Reviewer 1 Report

Unilateral vs bilateral CI users – why were these not analysed separately?

Why do the authors refer to patients and not clients? CI users are not sick people

Mean age and range of age varied for the study and control groups. Was this checked for statistical difference?

The subjects had different years of experience post-implantation. Why was this not controlled?

What alternative AT was the control group offered?

Was the ‘dedicated serious game’ validated before use?

Author Response

Unilateral vs bilateral CI users – why were these not analysed separately?

à Herein, the respective number of unilateral (n=9) vs bilateral CI subjects (n=4) in the study group, and unilateral (n=8) vs bilateral CI subjects (n=7) in the control group did not allow for separate analysis on that factor. We used a within-subject analysis to test for the efficacy of the serious game-based auditory training. Doing so, we found that all but one subject in the study group demonstrated SNR improvement post-training.

Why do the authors refer to patients and not clients? CI users are not sick people

We agree with that comment. Accordingly, the term “patients” has been replaced by “subjects” throughout the manuscript.

Mean age and range of age varied for the study and control groups. Was this checked for statistical difference? The subjects had different years of experience post-implantation. Why was this not controlled?

à Additional analyses have been conducted to take account of these remarks. Two paragraphs were added, respectively in the methods and results section. The discussion section has been refined accordingly in the revised manuscript.

Methods (page 7 line 225):

“To control for potential differences in demographics (age at testing, deafness duration prior to implant, years of implant experience) between groups, t-test and Wilcoxon test were used. Possible correlation between demographics and SRT70 improvement between T1 and T2 was also tested.”

Results (page 8 line 244):

“Mean age differed between the study and control groups (t-test, p=0.039). Age at testing, however, was not correlated with SRT70 improvement between T1 and T2 (Pearson test, p=0.525). Neither deafness duration prior to implant (t-test, p=0.449) nor the number of years of implant experience (Wilcoxon test, p=0.487) differed between groups. Further, SRT70 improvement between T1 and T2 did not correlate with deafness duration (Pearson test, p=0.071) nor with CI experience (Spearman test, p=0.360).”

Discussion (page 12 line 313):

“Among the participants’ demographic characteristics, only mean age differed between the study and control groups. None of the demographic characteristics, including deafness duration and experience with the implant, were found to correlate with improvement in SRT70.”

What alternative AT was the control group offered?

à Serious game based-auditory training was offered to subjects in the control group once the study was fully achieved. Identical audiological services were provided to all participants in our cochlear implant center.

Was the ‘dedicated serious game’ validated before use?

à The objective of the study was actually to validate the serious game in adult CI users through a 5-week personalized AT program. The serious game used was developed specifically for this purpose

Reviewer 2 Report

Comments:

The purpose of the study was to investigated the effects of a serious game for auditory training in speech-in-noise for post-lingually deafened adults. The description of the game is adequate as well as the descriptions of methodology and results.

Yet, several major flaws are noted in the study. The first is that the participants are not all post-lingually deafened but some have a prelingual hearing loss (see 1,3,7, and 15 in the study group). So, the participants do not correspond to the purpose of the study. Additionally there are differences in the mean age of the two groups and no statistical analysis took place to check whether there is a significant difference in the mean age across groups. Thirdly, the data in Table 2 Signal-to-noise ratio (dB) indicates that after the training (T2) the required signal to noise ration increase rather than decreases in several cases for the 70% discrimination score to be achieved. It should have been decreasing to show improvement. Fourth, the results in that part of Table 2 do not agree with the figures, as checked for different patients.

In sum, there are major flaws in the paper.

Author Response

The purpose of the study was to investigated the effects of a serious game for auditory training in speech-in-noise for post-lingually deafened adults. The description of the game is adequate as well as the descriptions of methodology and results. Yet, several major flaws are noted in the study. The first is that the participants are not all post-lingually deafened but some have a prelingual hearing loss (see 1,3,7, and 15 in the study group). So the participants do not correspond to the purpose of the study.

We apologize for this error. The study population included both pre- and post-language learning deaf subjects. This is now corrected in the manuscript to avoid confusion and to fully satisfy the stated purpose of the study. The term "post-lingual deafness" has been deleted and replaced with "bilateral profound deafness" throughout the manuscript.

Additionally there are differences in the mean age of the two groups and no statistical analysis took place to check whether there is a significant difference in the mean age across groups.

We thank the reviewer for this fruitful comment. Additional analyses have been conducted. Two paragraphs were added, respectively in the methods and results section. Also, the discussion section has been refined accordingly in the revised manuscript.

Methods (page 7 line 225):

“To control for potential differences in demographics (age at testing, deafness duration prior to implant, years of implant experience) between groups, t-test and Wilcoxon test were used. Possible correlation between demographics and SRT70 improvement between T1 and T2 was also tested.”

Results (page 8 line 244):

“Mean age differed between the study and control groups (t-test, p=0.039). Age at testing, however, was not correlated with SRT70 improvement between T1 and T2 (Pearson test, p=0.525). Neither deafness duration prior to implant (t-test, p=0.449) nor the number of years of implant experience (Wilcoxon test, p=0.487) differed between groups. Further, SRT70 improvement between T1 and T2 did not correlate with deafness duration (Pearson test, p=0.071) nor with CI experience (Spearman test, p=0.360).”

Discussion (page 12 line 313):

“Among the participants’ demographic characteristics, only mean age differed between the study and control groups. None of the demographic characteristics, including deafness duration and experience with the implant, were found to correlate with improvement in SRT70.”

Thirdly, the data in Table 2 Signal-to-noise ratio (dB) indicates that after the training (T2) the required signal to noise ration increase rather than decreases in several cases for the 70% discrimination score to be achieved. It should have been decreasing to show improvement.

--> After training, there was an increase in SNR in only one subject in the study group; in this subject, the increase was only 0.1 dB SNR. For the rest of the study group, a decrease in SNR was demonstrated. In contrast, SNR increased in 7 of 13 subjects in the control group.

Fourth, the results in that part of Table 2 do not agree with the figures, as checked for different patients.

--> We recognize that the use of hyphens in Table 2 may have caused confusion. To avoid misunderstanding, the improvement in SNR between T2 and T1, and T3 and T1 is now quoted as Delta T2 - T1 and Delta T3 - T1, respectively. Accordingly, negative Delta values reflect an improvement in SNR, and Figures 2 and 3 are in good agreement with the data in Table 2. 

Round 2

Reviewer 1 Report

The questions/comments were mostly addressed adequately.

With regards to validation of the tool, I was referring to other aspects of validity such as 'content validity'. This can be included  as a limitation of the study.

Author Response

Round 2 reviewing

Please find below our answer to the remaining question:

Reviewer 1:

The questions/comments were mostly addressed adequately.

With regards to validation of the tool, I was referring to other aspects of validity such as 'content validity'. This can be included  as a limitation of the study.

-->the reviewer refers to his previous question in round 1: “Was the ‘dedicated serious game’ validated before use?”

-->We replied in round 1: “The objective of the study was actually to validate the serious game in adult CI users through a 5-week personalized AT program. The serious game used was developed specifically for this purpose”

-->In a complementary way, to better answer the question, we have added a limit to the study in the "discussion" section.

  • Line 315
  • “The serious game we used was developed specifically for this study. The software and its content had not been subject to a previous validation study.”

-->we hope this clarification will meet the reviewer's expectations.

-->On behalf of the co-authors,

Yours sincerely